# Differentiation between Fresh and Thawed Cephalopods Using NIR Spectroscopy and Multivariate Data Analysis

**DOI:** 10.3390/foods10030528

**Published:** 2021-03-03

**Authors:** Francesco Pennisi, Alessandro Giraudo, Nicola Cavallini, Giovanna Esposito, Gabriele Merlo, Francesco Geobaldo, Pier Luigi Acutis, Marzia Pezzolato, Francesco Savorani, Elena Bozzetta

**Affiliations:** 1Istituto Zooprofilattico Sperimentale del Piemonte, Liguria e Valle d’Aosta, Via Bologna 148, 10154 Turin, Italy; francesco.pennisi@izsto.it (F.P.); giovanna.esposito@izsto.it (G.E.); pierluigi.acutis@izsto.it (P.L.A.); elena.bozzetta@izsto.it (E.B.); 2Department of Applied Science and Technology, Politecnico di Torino, Corso Duca degli Abruzzi 24, 10129 Turin, Italy; alessandro.giraudo@polito.it (A.G.); nicola.cavallini@polito.it (N.C.); francesco.geobaldo@polito.it (F.G.); francesco.savorani@polito.it (F.S.); 3Esselunga S.p.A, Via Giambologna 1, 20096 Limito di Pioltello, Milan, Italy; gabriele.merlo@esselunga.it

**Keywords:** NIR, cephalopods, freeze-thaw, chemometrics, food fraud, cuttlefish, musky octopus

## Abstract

The sale of frozen–thawed fish and fish products, labeled as fresh, is currently one of the most common and insidious commercial food frauds. For this reason, the demand of reliable tools to identify the storage conditions is increasing. The present study was performed on two species, commonly sold in large-scale distribution: Cuttlefish (*Sepia officinalis*) and musky octopus (*Eledone* spp.). Fifty fresh cephalopod specimens were analyzed at refrigeration temperature (2 ± 2 °C), then frozen at −20 °C for 10 days and finally thawed and analyzed again. The performance of three near-infrared (NIR) instruments in identifying storage conditions were compared: The benchtop NIR Multi Purpose Analyzer (MPA) by Bruker, the portable MicroNIR by VIAVI and the handheld NIR SCiO by Consumer Physics. All collected spectra were processed and analyzed with chemometric methods. The SCiO data were also analyzed using the analytical tools available in the online application provided by the manufacturer to evaluate its performance. NIR spectroscopy, coupled with chemometrics, allowed discriminating between fresh and thawed samples with high accuracy: Cuttlefish between 82.3–94.1%, musky octopus between 91.2–97.1%, global model between 86.8–95.6%. Results show how food frauds could be detected directly in the marketplace, through small, ultra-fast and simplified handheld devices, whereas official control laboratories could use benchtop analytical instruments, coupled with chemometric approaches, to develop accurate and validated methods, suitable for regulatory purposes.

## 1. Introduction

International food trade and the complexity of supply chains make nowadays the fight against food fraud a multifaceted and current challenge. The term “food fraud” refers to all those actions aimed at the production and/or distribution of food in violation of current legislation, intended to deceive the consumers, obtain illicit profits, and infringe the agri-food chain legislation. Fish products are among the most common defrauded food commodities [1], since they can be easily counterfeited and mislabeled [2,3,4,5]. One of the most occurring frauds concerns the declaration of freshness, which can be faked in case of frozen and thawed products. Specifically, selling frozen-thawed fish products labeled as fresh infringes the regulatory actions in Europe (European Parliament Legislative Resolution No 1169/2011 [6]) focused on providing the final consumer with information about the defrosted status of products by clear labeling. Moreover, it has been proven that multiple freezing–thawing cycles impair the wholesomeness of food by promoting the proliferation of bacteria, molds, and other microorganisms, especially in fish, thus significantly compromising the healthiness and safety related to its consumption [7,8].

For these reasons, the development of reliable analytical techniques, able to detect the fraudulent substitution of a fresh product with a thawed one, attracts, without a doubt, a great deal of the industrial research interest. In this context, different methods have been investigated over the past years, such as measurement of dielectric properties, erythrocyte and hematocrit evaluation, histology, protein extraction and proteomics, and enzymatic essays, [9,10,11,12,13,14], and they have proved to be effective for their applicability.

Conversely, all the above-mentioned techniques are rather time consuming and labor intensive, thus difficult to apply industrially due to the large amount of time required for sample treatment and to obtain the analysis results. Furthermore, the performance achieved to date are not those expected to assess food safety and quality.

As an attempt to overcome these issues, more rapid analytical techniques have been developed. Among them, the combined use of near-infrared (NIR) spectroscopy and chemometrics, has been suggested as a promising tool for building strategies to contrast food frauds: In fact, it allows differentiating the spectrum obtained from a fresh specimen, compared to one previously subjected to freezing and subsequently thawed [15].

Nowadays NIR spectroscopy is a well-established analytical technique, which has been already applied in quality control laboratories for monitoring the composition of intact samples (targeted analysis) of cereals, milk, and meat [16,17,18]. Moreover, NIR studies concerning both the freshness assessment of fish [15,19,20] and the fish authentication were reported, by also considering different species [21,22,23]. Surprisingly, to the best of authors’ knowledge, only few studies are available for rapid discrimination of fresh/thawed cephalopods by using NIR spectroscopy [24], even if they are a class of seafood highly exposed to fraudulent practices.

The Class *Cephalopoda*, and notably the Subclass *Coleoidea*, which includes a lot of taxa under the phylum of *Mollusca* (46 families, but classification is still under review), comprehends many species of high current interest to fisheries, such as squids, cuttlefishes, and octopuses [25]. Squids are by far the main component in the global cephalopod fishery production (about 70% of the total catch), followed by cuttlefishes (*Sepia* spp., *Sepiella* spp. and similar genera) and octopuses (mainly *Octopus* spp. and *Eledone* spp.). As an example, during the last 40 years, squid catches have increased by 250%, from a little more than 1 million tons in 1980 to over 3.6 million tons in 2014 [26]. The consumption of cephalopods meat is extensive and diversified: From fresh food, eaten raw as “sashimi” in Japan and worldwide, to fresh-cooked, and to many kinds of processed products (dried, frozen, canned, grounded, etc.). With the growing demand for food for human consumption, cephalopod resources are likely to receive even more attention in the future [25]. Furthermore, since the increase in attention and earnings often attracts illegal and malicious affairs, it is in everyone’s interest that good practices and innovative techniques are developed to prevent frauds and dishonesty.

The aim of the present study is indeed to set up an innovative and easy-to-use method to distinguish fresh from frozen–thawed cephalopod mollusks using NIR spectroscopy. Moreover, the use of handheld NIR instruments, which could be promising to quickly analyze a high number of samples directly on site, was considered. In detail, NIR spectra were acquired using three types of instruments: A benchtop instrument (Multi Purpose Analyzer or MPA, Bruker Optics Corporation), a portable medium/high-cost device (MicroNIR Pro Es, VIAVI Solutions) and a handheld, low-cost, and ultra-fast tool (SCiO, Consumer Physics). The evaluation of models’ accuracies in discrimination were performed for each type of instrument and critically compared. 

## 2. Materials and Methods

### 2.1. NIR Instruments (and Software)

Three NIR spectrometers with different spectral range and resolution were considered in this study: A benchtop Fourier transform–NIR (FT–NIR) spectrometer (Multi Purpose Analyzer–MPA, Bruker Optics, Ettlingen, Germany) equipped with a fiber optic reflectance probe, and two handheld instruments, namely the MicroNIR 1700 Pro ES (VIAVI Solutions, San Jose, CA, USA), and the SCiO Pocket molecular sensor (v1.2, Consumer Physics Inc., Tel Aviv, Israel). An overview of the technical features of the three instruments is given in (Table 1).

Analytical conditions for the MPA benchtop instrument were as follows: 800–2500 nm (12500–4000 cm^−1^) spectral range; 12 cm^−1^ resolution; 10 kHz scanner velocity, and 64 scans both for sample and background acquisition. Background scans were performed using instrumental internal reference standard. The *OPUS* software (v. 6.5, Bruker Optics, Ettlingen, Germany) was used for both instrumental control and spectra acquisition.

Analytical conditions for MicroNIR were as follows: 908–1676 nm spectral range, 12 cm^−1^resolution, 12.5 µs integration time, and 200 scans at 80 Hz were applied. The spectra acquisition and instrumental control were managed by means of the VIAVI *MicroNIR Pro* software (v2.0), operated on a laptop computer. 

Analytical conditions for the handheld SCiO device were as follows: 740–1070 nm (13514–9346 cm^−1^) spectral range, 10 cm^−1^ resolution, and a typical scan time of 2–5 s. Spectra collection and management was performed using the SCiO smartphone app (*The Lab*, version 1.3.1.81). The SCiO data management architecture is cloud-based; thus each collected spectrum is sent from the sensor to the smartphone via Bluetooth, and then uploaded and stored on the online *Consumer Physics Cloud* database. 

Instruments calibration were performed before the first acquisition and roughly every ten specimens, following the instruction of each NIR instrument.

### 2.2. Samples and Spectra Acquisition

A total of 50 fresh specimens of both *Sepia officinalis* (cuttlefish, hereafter also referred to as “SE”) and *Eledone* spp. (musky octopus, hereafter also referred to as “MO”) were collected straight from the processing factory of a local large-scale retailer. All samples were regularly fished and delivered to the processing factory, as a part of five distinct batches. These batches were collected on five different times between August 2019 and February 2020. Each specimen was analyzed without performing any kind of physical pre-treatment, using all the aforementioned NIR spectrometers, operating in diffuse reflectance mode.

The spectra acquisition using the two portable devices (SCiO and MicroNIR) were performed directly at the food factory; then, the same samples were transported, maintaining the cold chain (2 ± 2 °C), to the laboratory and acquired with the benchtop NIR spectrometer. Subsequentially, all samples were stored in controlled conditions at −20 °C, for 10 days, and subsequently thawed for 24 hours at 2 ± 2 °C to collect spectral data.

The spectral scans were conducted by examining the outer layer of the skin, according to a precise placement outline: 3 scans (i.e., “replicates”) were collected for each cuttlefish by scanning the lower mantle (or belly), which is paler and more homogeneous than the upper parts, and 3 scans (i.e., “replicates”) were acquired from the pale side of the head mantle of every musky octopus. 

For each sampling position one set of spectra was obtained and subsequently inspected and used for building a classification model. 

All the collected spectra were managed, pre-treated, pre-processed, and modeled as described in Section 2.3, where the statistical multivariate analysis methods and chemometric tools are described.

### 2.3. Statistical Analyses and Chemometric Methods

Each NIR instrument works with specific programs supplied by the manufacturer.

The spectra acquired with the MicroNIR are processed as raw data, exported in comma separated values (.*csv*) format and then organized in columns using *Microsoft Excel*. The SCiO raw data, stored in the *Consumer Physics Cloud*, were separately analyzed using chemometric approach and the analytical tools available in the online application provided by the manufacturer (*The Lab*, [27]). Finally, the MPA works with the *Opus Viewer* program (Bruker Corporation) and exports the spectra in *opus* format.

#### 2.3.1. Raw Data Pre-Treatment and Quality Assessment

The raw NIR spectral data acquired with the three instruments (Appendix A) were imported into MATLAB environment to be assessed, pre-processed, and modeled. The first and necessary step was the quality assessment, which was aimed at identifying potential faults in the data integrity as well as obvious outliers such as spectra with clear defects. The decision to eventually remove problematic samples was substantiated by assessing the groups of three replicates (as described in Section 2.2). The replicates analysis was performed by building a principal component analysis (PCA) (Section 2.3.2) model on the class-centered data i.e., after removing from each group of replicates its own mean spectrum. This approach allows building a model whose first principal component describes the deviation of each sample from the mean value of its group of replicates. By using the 95% confidence interval of the first principal component, it is possible to identify those samples which are significantly different from their group of replicates.

Once the problematic samples were identified, inspected, and finally removed, the average over each group of replicates was taken, generating a dataset with as many averaged spectra as the actual number of measured specimens i.e., one spectrum for each cephalopod. This pre-treatment procedure was applied to all three experimental datasets, therefore to each sample corresponded one spectrum in each one of the three final datasets used for modeling.

All datasets were pre-processed using standard normal variate (SNV) [28] to remove scatter effects.

Regarding SCiO raw data analyzed with the *The Lab* web application, they were modeled using four pre-defined algorithms (“Processed”, “Normalize”, “Processed & Normalize”, and “(log)R & Normalize”), which already include derivatives, SNV and a logarithmic transformation. No further information on the type of derivative is available since this is proprietary information. All collected data were considered suitable for the subsequent analysis and used to implement the classification models. The results obtained by the classification models were compared in terms of coefficient of variation (F_1_) and confusion matrix, with a pre-defined setting provided by the application.

#### 2.3.2. Exploratory Data Analysis

Principal component analysis (PCA, [29,30]) was used as an exploratory tool both for assessing the data quality (replicates analysis, Section 2.3.1) and to explore the pre-treated data before classification modeling. Datasets were pre-processed using SNV normalization followed by mean centering, prior to exploratory analysis.

#### 2.3.3. Classification Modeling

Partial least squares-discriminant analysis (PLS-DA, [31]) was used to discriminate between the classes of fresh and thawed specimens. Test set selection was performed using the Duplex algorithm [32] and it was applied on each class separately, to ensure a correct and balanced sampling of the frozen and thawed pools of samples.

Since the classification problem under examination only involves two classes (fresh vs. thawed), and a discriminant method is applied, the modeling results can be inspected from the point of view of one of the two classes: In this case, the prediction of the thawed samples class was chosen coherently with the aim of the study about fraud identification.

Four parameters were used to evaluate the models’ performances [31]:Specificity (Spec), which is the ability to avoid false positives i.e., fresh samples wrongly classified as thawed;Sensitivity (Sens), which is the ability to avoid false negatives i.e., thawed samples wrongly classified as fresh;Non-error rate (NER), which is computed as the mean of the sensitivities (one for each class) and corresponds to the model’s capability to correctly classify the samples;Accuracy (Acc), which is an estimation of the model’s error and is computed as the sum of the true positives (TP, correctly classified thawed samples) and true negatives (TN, correctly classified fresh samples) divided by the total number of samples (Equation (1)).(1)$$\mathrm{Accuracy}=\;\frac{\mathrm{TP}+\mathrm{TN}}{\mathrm{number}\;\mathrm{of}\;\mathrm{samples}}=\;\frac{\mathrm{correct}\;\left(\mathrm{thawed}\right)+\mathrm{correct}\;\left(\mathrm{fresh}\right)}{\mathrm{number}\;\mathrm{of}\;\mathrm{samples}}$$


All the models’ performance figures reported in the present article are referred to the point of view of predicting the thawed samples class, as previously stated. As a consequence of having two classes to model, the Spec and Sens values of the fresh class are going to be the “mirrored” version of the values corresponding to the thawed class: this means that the specificity of the thawed class is equal to the sensitivity of the fresh class, and vice versa.

The most influent variables of each classification model were inspected and identified using the “variable importance in projection” scores (VIP scores, [33]).

SCiO data modeled with *The Lab* web application were developed using the four algorithms, and thus the best model among the four obtained was chosen, according to coefficient of variation (F_1_) and confusion matrix. F_1_ defines the correlation between the spectra and the investigated parameter (storage conditions), while the confusion matrix defines the correspondence between the true class analyzed (fresh/thawed) and the class assigned by the model.

#### 2.3.4. Overview of the Exploratory and Classification Models

A total of six exploratory PCA models were built to inspect each dataset individually. Regarding the classification models, nine PLS-DA models were built:Three corresponding to cuttlefishes;Three corresponding to musky octopuses;Three “general” ones, corresponding to the union of cuttlefishes and musky octopuses.

These general models were built to test whether the fresh/thawed information shared between the cuttlefishes’ and musky octopuses’ data was robust enough to allow for good classification, even when the different cephalopod species are modeled together.

The same approach was used with data analyzed with SCiO *The Lab* web application, but in this case no “general” model was built. Models were only tested separately for each species, to discriminate between fresh/thawed specimens.

#### 2.3.5. Software and Toolboxes

The whole chemometric data analysis was carried out under MATLAB environment (The Mathworks, MA, USA, version 2017b). PCA exploratory analysis and PLS-DA classification modeling were performed using the functions included in the PLS_Toolbox (version 8.6, Eigenvector Research Inc. WA, USA) software package. Test set selection was performed using the Duplex function for MATLAB written by Michał Daszykowski [34]. In-house written routines were used to manage the raw data and organize the chemometric data analysis workflow.

Concerning the spectra analyzed with SCiO *The Lab* web application, details on software are as follows: Operated with *The Lab* smartphone app, version 1.3.1.81 (Consumer Physics Inc., Tel Aviv, Israel) installed on a personal Android phone; data storage in *Consumer Physics Cloud* (provided by VeriFood Ltd., Hod HaSharon, Israel); processing and modeling within *The Lab* web application environment [27].

## 3. Results

An overview of the information present in each dataset (six in total, two cephalopod species, and three analytical techniques), which was inspected using PCA, is given in Section 3.1: The PCA results are represented in (Figure 1). Then, the classification results regarding the cuttlefish (Section 3.2) and the musky octopus (Section 3.3) are reported. As anticipated in Section 2.3.4, a global classification model was also built, and its results are reported in Section 3.4. Finally, in Section 3.5, the SCiO results obtained using the analytical tools provided by the instrument’s manufacturer are shown.

### 3.1. PCA Exploratory Analysis Results

The purpose of a multivariate exploratory step is to gain knowledge about the information content of the data, from which it is possible to estimate spontaneous grouping in specimens’ clusters and how well a subsequent classification model could perform.

In this study it was found that all datasets, to different extents, contained some grouping tendencies (Figure 1). The musky octopus PCA results depict a clearer picture of the differences between fresh and thawed samples (Figure 1d–f), and this was confirmed by the PLS-DA modeling results, as reported in the lower section of (Table 2). However, the cuttlefish results, even if they appear less clear, still provide a good starting point for the following classification modeling, as the overlap between the fresh and thawed samples is not very extended (Figure 1a–c).

For the sake of brevity, the PCA loading plots, corresponding to the scores plots of (Figure 1), were reported as Appendix A.

### 3.2. Classification Results of the Cuttlefish’ (SE) Samples

The PLS-DA classification performance results for the cuttlefish’ samples are reported in the upper section of (Table 2). The best classification performances were obtained using the MicroNIR spectrometer, with which 94.1% of accuracy in prediction was obtained. The other two instruments provided significantly lower but still good classification performances: 85.3% of accuracy with the benchtop MPA and 82.3% of accuracy with the handheld SCiO device. The most influential wavelengths/wavenumbers, as represented in (Figure 2a–c) using the VIP scores, appear to be:SCiO, cuttlefish: Several individual peaks along the whole wavelength range;MicroNIR, cuttlefish: 1400–1550 nm and >1600 nm;MPA, cuttlefish: A group of peaks within the interval 1250–1667 nm (6000–8000 cm^−1^).

The cuttlefish SCiO model provided the lowest performances, and both the classification figures and the VIP scores representation (Figure 2a) suggest that this model may be less reliable than all the remaining eight models.

### 3.3. Classification Results of the Musky Octopus’ (MO) Samples

The PLS-DA classification performance results for the musky octopus’ samples are reported in the central section of (Table 2). Additionally, in this case, the best classification performances were obtained with the MicroNIR spectrometer, with 97.1% of accuracy in prediction. The performances obtained with the other two instruments are in this case not much lower, and the handheld SCiO sensor provided the second-best result, with 94.1% of accuracy in prediction. The benchtop MPA spectrometer follows with 91.2% of accuracy in prediction. The most influential wavelengths/wavenumbers, as represented in (Figure 2d–f) using the VIP scores, appear to be:SCiO, musky octopus: Two interesting groups of signals located around 950 nm and 1020 nm;MicroNIR, musky octopus: Mainly three groups of signals at 990 nm, 1350 nm and 1450 nm;MPA, musky octopus: Two groups of signals within the interval 5800–7500 cm^−1^ (1333–1724 nm).

### 3.4. Classification Results of the Global (SE + MO) Cephalopods Model for Fresh and Thawed Classification

The PLS-DA classification performance results of the global model are reported in the lower section of (Table 2). As for the previous cases, the MicroNIR data provided the best classification performances, with an accuracy of 95.6%. The MPA outcomes resulted almost equally good with 94.1% of accuracy, while the SCiO achieved 86.8% of accuracy. The most influential wavelengths/wavenumbers, as represented in (Figure 2g–i) using the VIP scores, appear to be:SCiO, global (SE + MO): Two interesting groups of signals located around 790 nm, 950 nm and 1020 nm;MicroNIR, global (SE + MO): Mainly three groups of signals at 990 nm, 1350 nm and 1450 nm;MPA, global (SE + MO): Two groups of signals within the interval 1333–1724 nm (5800–7500 cm^−1^).

### 3.5. Classification Results Obtained Using Pre-Defined Models in the SCiO the Lab Web Application

Besides the abovementioned elaboration, SCiO data were also analyzed using the four algorithms included in the web tool, to evaluate the capabilities of the classification without a systematic chemometric approach. The two species were analyzed separately, to discriminate between fresh from frozen–thawed specimens. No global model was performed, because in this case, the SCiO *The Lab* web application would include species variability and, therefore, it would become a four-classes model, whereas the interest of this work is only to evaluate the fresh/thawed discrimination performances. As far as it concerns cuttlefishes, “Processed and Normalize” algorithm was selected, after evaluation of coefficient of variation and confusion matrix. The best classification produced an F_1_ value of 0.846, a correct classification of 86% for fresh specimens (SE-f), and of 87% for frozen-thawed ones (SE-t), as shown in (Table 3) (accuracy: 86%). Better results were obtained in the case of musky octopuses, where the selected algorithm was “(log)R & Normalize”, and the best classification produced an F_1_ value of 0.939, with a 100% correct identification for fresh samples (MO-f), and 90% for frozen–thawed ones (MO-t), as shown in (Table 4) (accuracy: 95%). The algorithm for the validation process is proprietary and is, therefore, not accessible for users.

## 4. Discussion

As seen in the previous section, very promising classification results were achieved considering both the species and the kind of spectrometer used for data acquisition. In Section 4.1, a discussion concerning the PCA exploratory analysis is provided. Then, the PLS-DA classification models developed separately on cuttlefish, on musky octopus, and on both datasets, considered as global model, are discussed in Section 4.2. The Section 4.3 provides a brief interpretation of the spectral variables considered as most significant for the development of classification models.

Finally, in Section 4.4, SCiO results obtained using the instrument’s analysis tools are considered and discussed.

### 4.1. PCA Exploratory Analysis

Exploratory analysis is a cornerstone of the data analysis workflow, as it allows discovering groupings and trends present in the data, which may be used for further modeling, like the fresh/thawed classification models reported in this paper. However, it is important to consider that even if no clear separation between the frozen and thawed classes is recovered by PCA, the PLS-DA model could still be able to provide good classification results.

The PCA results, as shown in (Figure 1), depicted a rather clear picture of the information available for classification purposes. The musky octopus data seemed to yield clearer fresh/thawed separation (Figure 1d–f) compared to the cuttlefish data (Figure 1a–c) for all three analytical techniques. However, concerning the classification results, and therefore the extent to which the PLS-DA method was able to collect and process the discriminative information, this exploratory step represented a good starting point for the subsequent modeling steps.

### 4.2. PLS-DA Classification Models

#### 4.2.1. Cuttlefish (SE) Models

The results achieved by the classification analysis (i.e., PLS-DA) suggest that the model is able to distinguish the two product categories rather well. In fact, during the prediction of the external test set, only one sample was misclassified. In view of practical application, the main goal (or interest) is to be able in recognizing if a frozen product has been fraudulently sold as fresh, so the point of view of predicted thawed products has to be carefully considered.

Concerning the SE models developed with SCiO and MPA spectra, the achieved results were, in both cases, less performant than those achieved with the MicroNIR instrument. Actually, this aspect is not so clear and should be further investigated. The SCiO classification results substantially reflected the not good separation between F and T samples already observed by the exploratory data analysis, where the two classes appear quite overlapped among each other (Figure 1a). The ‘halfway’ performance of MPA could be ascribed to the more complex and comprehensive spectrum acquired by the instrument, which probably led to an excess of information, even potentially redundant. An explanation to the less performant results achieved by SCiO and MPA cuttlefishes’ models can also be derived from the inspection of the corresponding VIP scores plot (Figure 2), since too many non-negligible peaks can be found across the spectral window, with a crowded and rather noisy region between 833 and 1000 nm (12,000–10,000 cm^−1^) for the MPA spectral profile.

#### 4.2.2. Musky Octopus (MO) Models

The classification models developed on musky octopus led to very satisfying prediction results, confirming the clear trend towards separation between fresh and thawed samples already observed in PCA (Figure 1d–f). Moreover, similar performance was achieved, in this case, by all the three instruments. The prediction performed by the model developed on the MicroNIR spectra could be considered almost perfect, with just one wrongly assigned sample belonging to the external test set; SCiO performances were far better than those shown for cuttlefish prediction, and just two test set samples were misclassified. Concerning the MPA, as for cuttlefishes, perhaps the complexity of the spectral signal made it difficult to extract and elaborate the useful information, but still demonstrated remarkable prediction capability.

#### 4.2.3. Global Cephalopods (SE + MO) Models

The idea behind developing a global cephalopods model arose under the hypothesis that the two species investigated (SE and MO) have certain similarities, which are expected to be maintained also in the case of the fresh/thawed classification. As shown in (Table 2), the classification performances of the global model were comparable to those achieved for the single SE and MO models, when not even better. These results seem to prove the abovementioned hypothesis and could be attributed to a potential synergic behavior in the multivariate space of the two different datasets, which probably bring common information concerning the fresh/thawed distinction; information that is therefore strengthened by merging the SE and MO datasets.

### 4.3. VIP Scores Interpretation

The VIP scores plots reported in (Figure 2) suggest that the most important spectral variables selected by the algorithm for the global cephalopods model are quite consistent with those selected for the implementation of the individual SE and MO models. This aspect can especially be appreciated from the classification models related to the MicroNIR instrument.

The spectral region between 900 and 1400 nm is related to changes in the vibrational modes of O–H and C–H bonds [35,36]. This phenomenon can be essentially ascribed to proteolysis activity and relaxation of lipid structure during thawing. The release of exudates due to the potential rupture of the cell walls and protein denaturation affects, in turn, the scattering effect with the sample and the water interaction, in terms of ratio between free and bound water, which was found to be higher in thawed samples than in fresh ones [22].

The 1400–1500 nm wavelength range is related to the combination of O–H stretching and bending bands [37] and can be essentially linked to an increase of free water species during thawing [21,23,24]. 

Potential alteration of components such as lipids, proteins, amino, and fatty acids on thawed samples can be inferred by also looking at the wavelength’ absorption at 1600 nm, 1800 nm, and 2200 nm, since it originates from N–H first and second overtone, first overtone of CH aliphatic group stretching, and the combination of N–H and C=O bonds [36].

Despite the SCiO limitation, which substantially consists in covering a narrow range of wavebands compared to MicroNIR and MPA instruments, it was demonstrated how this simple device can be effectively used for the discrimination between fresh and thawed cephalopods samples, due to its ability to capture useful chemical information in the last part of visible and the first part of the near-infrared spectrum. Conversely, MPA instrument did not demonstrate to capture the same interesting information in the abovementioned range while, on the other hand, a very good signal in the water absorbance region could be appreciated, with a clear and consistent peak approximately at 1430 nm and, to a less extent, in the N–H and C–H absorbance region around 1670 nm. 

Finally, the MicroNIR instrument was found to represent, perhaps, the best solution among those investigated, since the generated spectral signal allowed gathering worthwhile and clear chemical information in the first part of the near-infrared spectrum but also in the shortwave infrared range between 1000 and 1600 nm. Therefore, this could be the key reason that, using this instrument, the best classification results were achieved for single SE and MO models and for the global cephalopods model as well.

### 4.4. Interpretation of SCiO Results Obtained Using Pre-Defined Models in the SCiO the Lab Web Application

By using a simple and rapid NIR device, it was possible to classify fish products according to storage conditions with remarkable performance, even though the SCiO *The Lab* web application only offers a limited number of statistical properties and it does not permit to freely elaborate data.

Nevertheless, almost all samples were correctly classified, and results obtained from the app were satisfactory even when some limitation, regarding data analysis, were reported. For MO model, the fresh specimens were never confused with the thawed ones (correct classification: 100%); whereas, for SE model, the correct classification was slightly minor (86%), probably due to the different size and homogeneity of the samples, but still good to help in detecting mislabeling of frozen–thawed products sold as fresh.

In conclusion, the results achieved from the SCiO sensor, using the pre-defined set of analysis, highlight the capabilities of this “user-friendly” device, to help even inexpert consumers in the field of food frauds. On the other hand, it does not allow to model many variables together, as previously specified regarding a “global model”, thus not making it easily applicable in official controls.

## 5. Conclusions

The purpose of the study was to investigate the accuracy of the three tools, and the additional goal was to set up a reliable model to be used in quality and official controls; at the same time, the results obtained with a ready-to-use device helped in verifying if this technique could be also disseminated among final consumers, to prevent commercial frauds. 

Selling frozen–thawed fish products, and in particular cephalopods, labeled as fresh, is one of the most occurring frauds because to date there is no reference analysis to identify with certainty their conservation status. NIR spectroscopy represents a quick and cost-effective technique and offers a series of undoubted advantages compared to traditional analytical methods, since it needs minimal or no sample preparation, it is non-destructive, and it does not entail heavy sample pre-treatments.

This study compared capabilities and performances of three different NIR techniques, for the discrimination between fresh vs. frozen–thawed cuttlefishes (*Sepia officinalis*) and musky octopuses (*Eledone* spp.), as they are among the most relevant species in the global cephalopod fishery production.

All the investigated techniques showed good accuracy in the classification of fish products, therefore NIR spectroscopy proved to be a powerful technology in the fight against food frauds.

Moreover, it is important to underline that the best results were obtained using handheld and portable tools, which greatly reduces the instrumental complexity and simplifies the execution of the analysis, making it much more practical and suitable for screening directly on the production line.

Based on these findings, food frauds could be detected directly in the marketplace by consumers and companies through small portable devices with pre-set analyses, whereas official control laboratories would rely on analytical instruments coupled with chemometric approaches, in order to develop accurate and validated methods, suitable for regulatory purposes.

## Figures and Tables

**Figure 1 foods-10-00528-f001:**
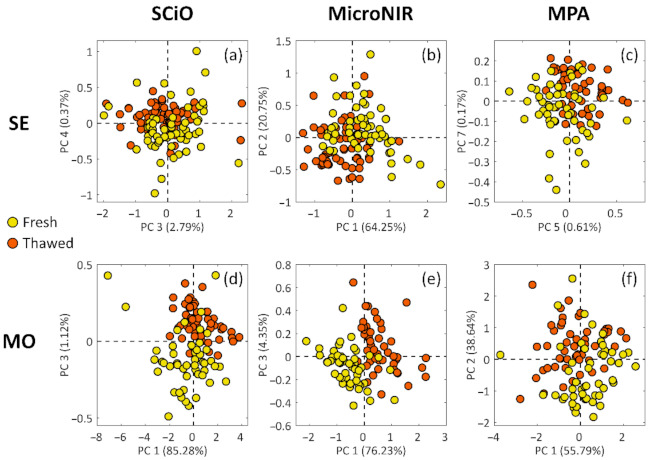
Most relevant PCA scores plots of cuttlefish’ (SE; (**a**–**c**)) and musky octopus’ (MO; (**d**–**f**)) samples. Fresh samples are depicted in yellow, while thawed samples are colored in orange.

**Figure 2 foods-10-00528-f002:**
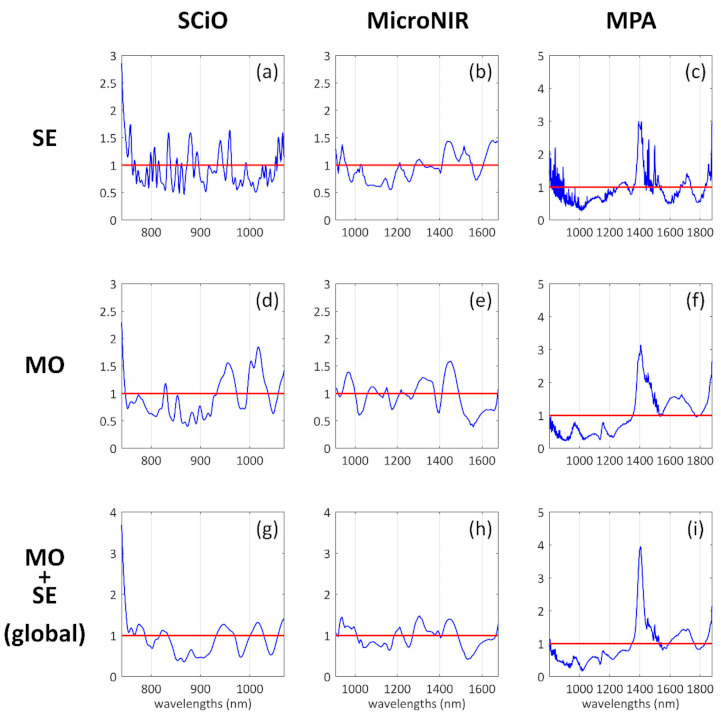
Partial least squares-discriminant analysis (PLS-DA) variable importance in projection (VIP) scores of cuttlefish’ (SE; (**a**–**c**)), musky octopus’ (MO; (**d**–**f**)) and global (**g**–**i**) models.

**Table 1 foods-10-00528-t001:** List of the main technical near-infrared (NIR) spectrometers’ characteristics.

	Size (cm, W × L × H)	Weight (g)	Cost ($)	Spectral Range (nm)
MPA (Bruker)	37.5 × 59.3 × 26.2	3500	≈150,000	800–2500
MicroNIR (VIAVI)	4.6 × 4.6 × 5	250	≈35,000	908–1676
SCiO (Consumer Physics)	1.5 × 4 × 6.5	<50	<5000	740–1070

**Table 2 foods-10-00528-t002:** Classification results: fresh/thawed cuttlefishes (SE) and musky octopuses (MO). All values are expressed in percentage.

		SCiO					MicroNIR					MPA				
		LVs	Spec	Sens	NER	Acc	LVs	Spec	Sens	NER	Acc	LVs	Spec	Sens	NER	Acc
SE	Cal		97.0	87.9	92.4	92.4		97.0	100	98.5	98.5		100	97.0	98.5	98.5
	CV	9	84.8	69.7	77.3	77.3	6	97.0	97.0	97.0	97.0	8	87.9	81.8	84.8	84.8
	Test		70.6	94.1	82.3	82.3		94.1	94.1	94.1	94.1		82.3	88.2	85.3	85.3
MO	Cal		97.0	97.0	97.0	97.0		100	100	100	100		97.0	93.9	95.4	95.4
	CV	9	97.0	93.9	95.4	95.4	7	100	100	100	100	6	93.9	90.9	92.4	92.4
	Test		94.1	94.1	94.1	94.1		94.1	100	97.1	97.1		88.4	94.1	91.2	91.2
global	Cal		93.9	75.8	84.8	84.8		98.5	95.4	97.0	97.0		90.9	84.8	87.9	87.9
	CV	6	95.4	71.2	83.3	83.3	5	97.0	93.9	95.4	95.4	5	87.9	81.8	84.8	84.8
	Test		85.3	88.2	86.8	86.8		97.1	94.1	95.6	95.6		91.2	97.1	94.1	94.1

**Table 3 foods-10-00528-t003:** Confusion matrix for cuttlefishes: SE-t (thawed); SE-f (fresh). Performance: F_1_ = 0.846. All values are expressed in percentage.

		Known Class	
		SE-t	SE-f
**Classified**	**SE-t**	87	14
	**SE-f**	13	86

**Table 4 foods-10-00528-t004:** Confusion matrix for musky octopuses: MO-t (thawed); MO-f (fresh). Performance: F_1_ = 0.939. All values are expressed in percentage.

		Known Class	
		MO-t	MO-f
**Classified**	**MO-t**	90	0
	**MO-f**	10	100

## Data Availability

Data available on request.

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
