# Peer review of "Differentiation between Fresh and Thawed Cephalopods Using NIR Spectroscopy and Multivariate Data Analysis"

_foods, 2021, doi:10.3390/foods10030528_

Round 1

Reviewer 1 Report

Using three types of NIR spectrometers and chemometric analysis including PCA, and PLS-DA, the authors discriminated the difference between fresh and thawed samples for screening the food frauds. Besides, they evaluated the applicability of a handheld, portable spectrometer by investigating their performance. 
Overall, it looks like a well-written and well-organized paper, but it would be better to add a few things as follows:

In the conclusion section, the purpose of the study was revealed, namely, to examine the performance like the accuracy of three different spectrometers. And the outstanding results of the portable spectrometers were mentioned. However, it would be better if the motivation for using the handheld spectrometers could be more addressed in the introduction. And it needs to be emphasized more in the abstract, relating to their results.

Furthermore, since some technical descriptions of the spectrometer have not been dealt with efficiently, it seems necessary to provide more information by comparing their performance specifications (cost, weight, size, etc.) and structures on one table. 

Lastly, it is a minor part, but it would be better to unify the horizontal axis dimension in Figure 2 with the wavelength scale.
Unappropriate capitalization (or missing abbreviations) in some sentences should be corrected. (L120, L155, L184, L211)

Author Response

Please see the attachment  for figures.

Manuscript ID: foods-1110827

Title: Differentiation between Fresh and Thawed Cephalopods Using NIR Spectroscopy and Multivariate Data Analysis

We thank the reviewers for their helpful comments and the careful review of our manuscript and editor for the swift processing time.

In this letter, as suggested, reviewers’ comments are addressed point–by-point and reported in Italics.

In the revised manuscript the changes are tracked.

Graphical abstract and supplementary materials are also provided.

Thank you in advance for your kind attention

Best regards

REVIEWER 1

Using three types of NIR spectrometers and chemometric analysis including PCA, and PLS-DA, the authors discriminated the difference between fresh and thawed samples for screening the food frauds. Besides, they evaluated the applicability of a handheld, portable spectrometer by investigating their performance.

Overall, it looks like a well-written and well-organized paper, but it would be better to add a few things as follows:

In the conclusion section, the purpose of the study was revealed, namely, to examine the performance like the accuracy of three different spectrometers. And the outstanding results of the portable spectrometers were mentioned.

However, it would be better if the motivation for using the handheld spectrometers could be more addressed in the introduction.

REPLY: we would like to thank the reviewer for the precious insight. Authors rephrased the aim of the study accordingly to better explain the motivation for using the handheld spectrometers.

Line 90-98: “...Aim of the present study is indeed to set up an innovative and easy-to-use method to distinguish fresh from frozen-thawed cephalopod mollusks using NIR spectroscopy. Moreover, the use of handheld NIR instruments, which could be promising to quickly analyze a high number of samples directly on site, was considered. In details, NIR spectra were acquired using three types of instruments: a benchtop instrument (MPA, Bruker Optics Corporation), a portable medium/high-cost device (MicroNIR Pro Es, VIAVI Solutions) and a handheld, low-cost and ultra-fast tool (SCiO, Consumer Physics). The evaluation of models’ accuracies in discrimination were performed for each type of instrument and critically compared

And it needs to be emphasized more in the abstract, relating to their results.

REPLY: once again, we would like to thank the reviewer for the precious insight. Authors rephrased the final part of the abstract accordingly, in order to emphasize the results obtained using portable spectrometers and their direct applicability in field to fight against food frauds.

Line 30-33.“….Results show how food frauds could be detected directly in the marketplace, through small, ultra-fast and simplified handheld devices, whereas official control laboratories could use benchtop analytical instruments, coupled with chemometric approaches, to develop accurate and validated methods, suitable for regulatory purposes.”

 Furthermore, since some technical descriptions of the spectrometer have not been dealt with efficiently, it seems necessary to provide more information by comparing their performance specifications (cost, weight, size, etc.) and structures on one table.

We would like to thank the reviewer for the helpful comment. The authors structured the main technical specifications of the 3 different NIR instruments in a table (Table 1), which was added in Section 2.1 together with a brief reference to it in the 1st paragraph of this section: “An overview of the technical features of the three instruments is given in Table 1”.

Size (cm, W × L × H)

Weight (g)

Cost ($)

 Spectral range (nm)

MPA Bruker

37.5 × 59.3 × 26.2

3500

≈ 150,000

800 – 2500

MicroNIR Viavi

4.6 × 4.6 × 5

250

≈ 35,000

908 – 1676

NIR SCiO Consumer Physics

1.5 × 4 × 6.5

<50

< 5000

740 – 1070

Table 1: List of the main technical NIR-spectrometers’ characteristics.

 Lastly, it is a minor part, but it would be better to unify the horizontal axis dimension in Figure 2 with the wavelength scale.

We would like to thank the reviewer for the valuable suggestion. Authors have modified the horizontal axis in Figure 2 accordingly.

Figure 2 (same caption as before…)

 Unappropriate capitalization (or missing abbreviations) in some sentences should be corrected (L120, L155, L184, L211)

We would like to thank the reviewer for the useful remarks. Authors provided the following changes along the manuscript:

L131 changed to “large-scale retailer”

L168 changed to “principal component analysis (PCA)”

L181 changed to “standard normal variate (SNV)”

L192 changed toPrincipal component analysis”

L197 changed to “Partial least squares-discriminant analysis (PLS-DA)

Reviewer 2 Report

The manuscript provides an interesting and simple concept of fresh and thawed octopus identification. The significance of this method lies primarily in its simple introduction into industry, especially when using handheld devices. However, this article does not address this significance very much and should be taken only as proof of concept.

Only two comments:

  • example of original (raw) spectra of fresh and thawed SE or MO should be provided for all three instruments (e.g. in supplement). Statistical method works with whole spectral interval, but it is not clear, which parts of spectra are significant and how complex the spectra are.
  • the significance or insignificance of individual spectral bands should be proved by PCA-loadings chart with wavenumber (cm-1) on x-axis (e.g. also in supplement). This provide a experimental support for statements in 4.3 (row 405-432) where the types of compounds and their vibrational modes are described.

Author Response

Please see the attachment for figures.

Manuscript ID: foods-1110827

Title: Differentiation between Fresh and Thawed Cephalopods Using NIR Spectroscopy and Multivariate Data Analysis

We thank the reviewers for their helpful comments and the careful review of our manuscript and editor for the swift processing time.

In this letter, as suggested, reviewers’ comments are addressed point–by-point and reported in Italics.

In the revised manuscript the changes are tracked.

Graphical abstract and supplementary materials are also provided.

Thank you in advance for your kind attention

Best regards

REVIEWER 2

The manuscript provides an interesting and simple concept of fresh and thawed octopus identification. The significance of this method lies primarily in its simple introduction into industry, especially when using handheld devices. However, this article does not address this significance very much and should be taken only as proof of concept.

Only two comments:

  • example of original (raw) spectra of fresh and thawed SE or MO should be provided for all three instruments (e.g. in supplement). Statistical method works with whole spectral interval, but it is not clear, which parts of spectra are significant and how complex the spectra are.

We would like to thank the reviewer for the valuable suggestion. The representation of original (raw) NIR spectra acquired on both fresh and thawed SE and MO using the three NIR instruments has been provided as supplementary material (Figure 1S) with a brief reference to it in the 1st paragraph of Section 2.3.1: “The raw NIR spectral data acquired with the three instruments (Figure 1S) […]”.

 FIGURE 1s CAPTION

Figure 1S. Example of raw NIR spectra of cuttlefish (“SE”; a, b, c) and musky octopus (“MO”; d, e, f) samples acquired using SCiO, MicroNIR and MPA instruments. In yellow are depicted fresh samples, while thawed samples are colored in orange.

  • the significance or insignificance of individual spectral bands should be proved by PCA-loadings chart with wavenumber (cm-1) on x-axis (e.g. also in supplement). This provide a experimental support for statements in 4.3 (row 405-432) where the types of compounds and their vibrational modes are described.

We would like to thank the reviewer for the interesting remark. However, we would argue that, from a chemometric viewpoint, the significance (or “importance”) of the individual spectral bands should always be discussed in relation to the classification results obtained by PLS-DA modeling, and not straight on the basis of the PCA model results. Indeed, PCA just gives an idea about the potential sample clustering in a completely unsupervised way, while the information about the variables, which are important for a specific purpose, in this case the discrimination between fresh and thawed samples, can be obtained from the supervised classification models. The abovementioned information is summarized in the VIP (variable importance in projection) scores, whose behavior was graphically represented in Figure 2. All the statements reported in Section 4.3 (row 423-458) were thus directly derived from the interpretation of the important bands identified in Figure 2.

Nevertheless, we have provided the PCA loading plots as supplementary material (Figure 2S) in order to strengthen the information conveyed by the PCA exploratory analysis. A brief reference to it was added in the last paragraph of Section 3.1: “For the sake of brevity, the PCA loading plots corresponding to the scores plots of Figure 1 were reported as supplementary material (Figure 2S).”

 FIGURE 2s CAPTION 

Figure 2S. PCA loadings corresponding to the principal components (PC) shown in Figure 1 (PCA score plots). Models corresponding to cuttlefish (“SE”; a, b, c) and musky octopus (“MO”; d, e, f) individual datasets are reported.
